# Surveillance of Antimicrobial Prescriptions in Community Pharmacies Located in Tokyo, Japan

**DOI:** 10.3390/antibiotics12081325

**Published:** 2023-08-17

**Authors:** Kosuke Hasegawa, Tomoko Mori, Toshio Asakura, Yuriko Matsumura, Hidemasa Nakaminami

**Affiliations:** 1Department of Clinical Microbiology, School of Pharmacy, Tokyo University of Pharmacy and Life Sciences, 1432-1 Horinouchi, Hachioji, Tokyo 192-0392, Japan; hasegawa.k@mdx-gr.com; 2MEDIX, Inc. 1-2-3 Motoyokoyamacho, Hachioji, Tokyo 192-0063, Japan; mori.t@mdx-gr.com (T.M.); asakura.t@mdx-gr.com (T.A.); matsumura.y@mdx-gr.com (Y.M.); 3Shinwa Pharmacy Minamishincho Store, 13-21 Minamishincho, Hachioji, Tokyo 192-0075, Japan

**Keywords:** AMR Action Plan, antimicrobial prescription, DPY, pharmacy, antimicrobial stewardship policy in pediatric clinic

## Abstract

An antimicrobial resistance (AMR) Action Plan was launched in 2016 to prevent the spread of antimicrobial-resistant bacteria in Japan. Additional support for the appropriate use of pediatric antimicrobial agents was initiated in 2018 to promote the appropriate use of antimicrobial agents in the community. To evaluate the effectiveness of the AMR Action Plan in the community, we investigated antimicrobial prescriptions in community pharmacies. Data on prescriptions for antimicrobial agents dispensed in 42 community pharmacies located in the Tama district, Tokyo, Japan, were collected between April 2013 and December 2019. In this study, we employed the DPY, which was calculated as defined daily doses (DDDs)/1000 prescriptions/year. The DPY is the number of antimicrobial agents used (potency) per 1000 antimicrobial prescriptions dispensed in pharmacies per year. The number of prescriptions for third-generation cephalosporins, fluoroquinolones, and macrolides decreased after the initiation of the AMR Action Plan; the DPYs of these antimicrobial agents decreased significantly by 31.4%, increased by 15.8%, and decreased by 23.6%, respectively (*p* < 0.05). The number of antimicrobial prescriptions for pediatric patients has been decreasing since 2018. Declines in the DPYs of third-generation cephalosporins, fluoroquinolones, and macrolides were higher in pediatric pharmacies than in other pharmacies. Our data suggest that the AMR Action Plan and additional support for the appropriate use of antimicrobial agents in children influenced the number of antimicrobial prescriptions in community pharmacies in Japan.

## 1. Introduction

In Japan, the antimicrobial resistance (AMR) Action Plan was launched in 2016 [1]. The main objectives of the plan were to reduce overall antimicrobial use by 33%; reduce the use of oral third-generation cephalosporins, fluoroquinolones, and macrolides by 50%; and reduce the use of injectable antimicrobials by 20% until 2020. According to the Nippon AMR One Health Report 2021, antimicrobial sales decreased by 29.9% compared to the sales in 2013 [2]. In particular, the sales of cephalosporins, fluoroquinolones, and macrolides decreased to 42.7%, 41.3%, and 39.3%, respectively, in 2020. However, the proportion (48.1%) of methicillin-resistant *S. aureus* (MRSA) in 2020 remained the same (47.7%) as that in 2016. Thus, although the use of antimicrobial agents decreased, no change in the prevalence of MRSA has been observed. One of the reasons for this was the increased prevalence of community-acquired MRSA in hospitals [3,4].

Antimicrobial usage among outpatients in Japan is higher than that in other countries [5]. For more than half of the outpatients, antimicrobial prescriptions are considered unnecessary, and antimicrobial agents that are not first-line agents are occasionally prescribed [6,7]. If the use of antimicrobials for outpatients is not improved, resistant strains of bacteria will emerge. Although antibiotic use is lower in Japan than in European countries, the proportional use of broad-spectrum antibiotics (such as cephalosporins, fluoroquinolones, and macrolides) is higher [8]. A previous report suggested that the overuse of broad-spectrum antibiotics can promote antimicrobial resistance and lead to adverse effects [9]. Thus, broad-spectrum antibiotics should be used sparingly [8]. A previous study reported that the use of antimicrobial agents was reduced by refining the prescription practices of physicians with the cooperation of medical associations; however, it has been suggested that the effect weakens over time [10]. The promotion of antimicrobial stewardship for outpatients is a vital part of the AMR Action Plan [11]. To effectively control AMR, it is necessary to have a detailed understanding of the community situation and to have process indicators, such as antimicrobial use, so that pharmacists can intervene and dispense antibiotics judiciously at pharmacies. To evaluate the promotion of appropriate antimicrobial use by pharmacists working in community pharmacies, it is necessary to identify trends in antimicrobial use in each pharmacy and the characteristics of antimicrobial use according to the institutional setting [12]. In this context, additional measures were taken to support the appropriate use of pediatric antimicrobial agents in 2018. Meeting the requirements for the evaluation of medication, including explaining to physicians that antimicrobial agents need not always be prescribed, can ensure rational antibiotic usage. Evaluations were performed for infants under three years of age from April 2018 to 31 March 2020. From 1 April 2020, infants up to six years of age were covered in the study. To date, many antimicrobial drug studies have been conducted based on antimicrobial sales data from wholesalers and antimicrobial data from insurance claims received [5,6,13,14]. In the case of sales data obtained from wholesalers, antimicrobials purchased by medical institutions may remain in the inventory. Although studies examining antimicrobial use nationwide have been reported, there are no studies examining antimicrobial use in specific regions. There are still scattered cases of inappropriate antimicrobial prescriptions and cases of the AMR Action Plan being insufficiently implemented in local clinics and pharmacies [11]. Limited studies have focused on actual antimicrobial prescriptions dispensed by pharmacies. Studies have examined the intermittent changes in the quantity of antibiotics used between January 2019 and June 2020 [12]. However, these were based on intermittent usage data and did not portray data over a continuous period. Although previous studies have calculated the DPM based on monthly prescriptions, the current study was based on annual prescriptions and defined the DPY to assess continuous usage. To the best of our knowledge, this is the first study that examines data on the dispensation of antimicrobial prescription in a specific geographical region, and only a few studies have validated the impact of an additional fee for the appropriate use of pediatric antimicrobial agents established in April 2018.

Here, we examined annual antimicrobial prescription data based on prescriptions dispensed at community pharmacies to evaluate the AMR Action Plan in the community.

## 2. Materials and Methods

### 2.1. Analysis of Prescription Data

The study protocols were approved by the Pharmacy Society of Japan Ethics Committee (20005) and the Tokyo University of Pharmacy and Life Sciences Ethics Committee (20-9). Prescription details were tabulated using only the date the antibiotic was dispensed, age, gender, and drug name to protect the identities of individuals. No personally identifiable information, such as drug history, was provided.

Data on antimicrobial prescriptions from 42 community pharmacies located in the Tama district (27, 13, and 2 in the South, North, and West Tama areas, respectively), Tokyo, Japan, were collected between April 2013 and December 2019. The data consisted of the dispensing date, the patient’s age and sex, the name of antimicrobial agents, and the number of prescriptions. Furthermore, antimicrobial prescriptions for children under three years of age in 12 pharmacies located in front of pediatric clinics were analyzed to investigate the impact of the additional support provided for the appropriate use of pediatric antimicrobial agents. A total of 556 insurance pharmacies were found to be located in the study area. The population of Tokyo totaled 13,960,236 in January 2021. Among them, 3,032,913 people (21.7%) were found to live in the area of this study. We also found that the percentage of the population aged 65 and over in the area is 25.5%, which is slightly higher than the average (22.6%) in Tokyo [15]. We could not take into account people moving in or out of the area. This should be mentioned as a limitation of this study.

This study only included oral antimicrobials and excluded injections, topical antimicrobials, antifungals, and antiparasitic agents. The classification of antimicrobial agents was based on the Anatomical Therapeutic Chemical (ATC) classification system (Appendix A) [16]. The results were displayed using the AWaRe (Access, Watch, Reserve) classification (https://aware.essentialmeds.org/list) (accessed on 8 August 2023).

### 2.2. Calculation of Antimicrobial Usage

A common index of antimicrobial use, DID, was calculated as the defined daily doses (DDDs)/1000 persons/day [17]. The DID represents the daily usage of 1000 patients by usage (potency) and is primarily a unit of measure of usage in a region or country. In this study, we employed the DPY, which was calculated as DDDs/1000 prescriptions/year. The DPY is the number of antimicrobial agents used (potency) per 1,000 antimicrobial prescriptions dispensed in pharmacies per year. Because it is based on prescriptions actually dispensed in pharmacies, it is a unit that indicates the number of antimicrobial agents used on a regional or store-by-store basis. The DPY is expressed in terms of usage per year for 1000 prescriptions and is based on actual prescriptions dispensed at pharmacies. Thus, DPY is considered more suitable than DID for evaluating antimicrobial use in pharmacies. The Drug Use Visualization Assisting Tool (DUVAT) was used to calculate the DDDs, and the accompanying Microsoft Excel (Microsoft Corporation, WA, USA) file was used to calculate the DPY [18].

The DPYs were calculated for 17 representative pharmacies with a high number of accepted prescriptions, excluding 5 pharmacies located in front of pediatric clinics. To determine the impact of the addition of appropriate pediatric antimicrobial prescriptions for children under three years of age, 12 pediatric pharmacies were tabulated. Of the 12 pharmacies, the DPYs were calculated for the five gate pharmacies that were exclusively pediatric pharmacies. 

### 2.3. Statistical Analysis

Differences in the number of antimicrobial prescription rates and DPYs were tested using the Mann–Whitney *U* test. Statistical significance was set at *p* < 0.05.

## 3. Results

### 3.1. Annual Transition of Antimicrobial Prescription in Community Pharmacies

The annual transitions in antimicrobial prescriptions in 42 pharmacies from April 2013 to 2019 were analyzed (Table 1 and Appendix A). The number of antimicrobial prescriptions decreased by 23.5% from 2014–2016 (260,768) to 2017–2019 (199,556). In contrast, the overall number of prescriptions, excluding antimicrobial agents, during the study period remained almost unchanged: 179,205 in 2013 (from April); 250,106 in 2014; 252,415 in 2015; 252,737 in 2016; 253,251 in 2017; 264,378 in 2018; and 267,573 in 2019. The percentage of prescriptions for antimicrobial agents during the study period was approximately 10%. Therefore, the decrease in the number of antimicrobial prescriptions was not due to a change in the number of prescriptions accepted by pharmacies. Third-generation cephalosporins (209,080; 39.6%), fluoroquinolones (104,474; 19.8%), and macrolides (137,962; 26.2%) accounted for 89.8% of all prescriptions (Table 1). Additionally, the total number of antimicrobial prescriptions decreased from 2016, i.e., the year when the AMR Action Plan was initiated. The total number of prescriptions for third-generation cephalosporins, fluoroquinolones, and macrolides significantly decreased to 30.6% (from 106,445 to 73,893), 10.3% (from 48,714 to 43,717), and 31.6% (from 71,078 to 48,592), respectively, between 2014–2016 and 2017–2019 (*p* < 0.05). Overall, the frequency of antimicrobial prescriptions decreased; however, the use of third-generation cephalosporins, fluoroquinolones, and macrolides remained high (Table 1). The use of fluoroquinolones and combinations of sulfonamides and trimethoprim, including derivatives that accounted for the total number of prescriptions, increased annually (Appendix A). In contrast, the proportion of third-generation cephalosporin and macrolide prescriptions decreased annually. The number of prescriptions for fluoroquinolones itself already decreased, but their share of total prescriptions increased (Appendix A).

### 3.2. Impact of the Additional Support for the Appropriate Use of Pediatric Antimicrobial Agents

The annual transitions of antimicrobial prescriptions for infants and young children in 12 pharmacies located in front of pediatric clinics were investigated (Figure 1 and Appendix A). The total number of antimicrobial prescriptions for children under three years of age at pediatric pharmacies has decreased since 2016 and has continued to decrease since 2018 (the year when additional support for the appropriate use of pediatric antimicrobials was implemented) (Figure 1). The number of extended-spectrum penicillin prescriptions increased once in 2018 but declined in 2019. However, the number of prescriptions of third-generation cephalosporins and macrolides decreased annually. The proportion of fluoroquinolone prescriptions rose until 2017, despite the publication of the AMR Action Plan, but declined since 2018 (Appendix A).

### 3.3. The DPY of Antimicrobial Agents

Compared to the DPYs between 2014–2016 and 2017–2019, the DPY of third-generation cephalosporins and macrolides decreased by 31.4% and 23.6%, respectively (*p* < 0.05). However, the DPY for fluoroquinolones increased by 15.8% (*p* < 0.05) (Table 2, Appendix A).

During the same period, the DPY in the five pediatric pharmacies decreased by 37.4%, 26.8%, and 41.3% for third-generation cephalosporins, fluoroquinolones, and macrolides, respectively (*p* < 0.05) (Table 3, Appendix A). The data showed that the decrease in the DPYs of oral third-generation cephalosporins, fluoroquinolones, and macrolides was higher in pediatric pharmacies than in other pharmacies.

## 4. Discussion

The present study showed the potential effects of the AMR Action Plan on reducing the number of antimicrobial prescriptions in 42 community pharmacies in the Tama area of Tokyo. To the best of our knowledge, this is the first antimicrobial prescribing study conducted in a limited number of community pharmacies. In this study, we compared the AMR Action Plan in the three years before and after to see its impact. Our data clearly showed a downward trend in the number of antimicrobial prescriptions since 2016. In addition, this study was not affected by the COVID-19 pandemic, as data were collected through December 2019, before the pandemic began in Japan. This study can serve as a guideline for understanding the efficiency of the AMR Action Plan, thereby facilitating the improved implementation of the plan in Japanese community pharmacies, and promoting more suitable infection control measures and prescription practices. A nationwide study of antimicrobial prescriptions in pharmacies between 2019 and 2021 reported that the use of third-generation cephalosporins, quinolones, and macrolides in pharmacies that received prescriptions primarily from hospitals or clinics decreased [12]. However, the previous report did not cover the AMR Action Plan or the impact of the addition of the pediatric antimicrobial regimen on the appropriate use of antimicrobials. To the best of our knowledge, this is the first report on antimicrobial prescriptions in pharmacies that focuses on the appropriate use of pediatric antimicrobials. The results for quinolone antimicrobials also differed from those in this study. The previous study included pharmacies across the country. This suggests that antimicrobial use may vary by region. Therefore, it was suggested that it is necessary to promote the appropriate use of antimicrobial agents after understanding the situation of antimicrobial use in each region.

The outcomes of the AMR Action Plan are as follows: an overall decrease of 28.9% in the number of antimicrobial prescriptions and decreases of 42.8%, 39.5%, and 41.5% in the use of third-generation cephalosporins, macrolides, and fluoroquinolones, respectively [11]. In contrast, our data showed that the total number of antimicrobial prescriptions decreased by 23.5%, the use of third-generation cephalosporins and macrolides decreased by approximately 30%, and the use of fluoroquinolones decreased by approximately 10%. The results of this study did not meet the AMR Action Plan targets. To achieve this outcome measure, community pharmacists should actively intervene in antimicrobial prescribing and promote the appropriate use of antimicrobials. Interestingly, an increased DPY was observed for fluoroquinolone prescriptions among all antimicrobial agents, possibly due to the increase in DDD due to the prolonged number of days of administration. Even if the number of antimicrobial prescriptions decreases, the DPY increases as the number of prescription days increases, suggesting the need for the further review of antimicrobial usage and other factors. One possible reason for the increase in DPYs for fluoroquinolones is the increase in DDDs due to longer days of administration. Even if the number of prescriptions for antimicrobial agents is decreasing, if the number of days prescribed increases, the DPY will also increase, suggesting the need for further review of antimicrobial usage and other factors. Additionally, this may have been due to the high frequency of fluoroquinolone prescriptions. Fluoroquinolones require fewer doses per day than other antimicrobial agents and have a broader spectrum of antimicrobial activity, which makes physicians prescribe this antibiotic. This may be the reason for the rate of decline in use to be in the lower 10% range and may also explain an increase in the DPY for this antibiotic. To further reduce antimicrobial use, pharmacists must intervene against unnecessary antimicrobial prescriptions and provide appropriate information to physicians and patients. Therefore, there is an urgent need to address not only children but also adults and older adults. In contrast, the prescription rate of penicillin has increased annually presumably due to the AMR Action Plan which promotes the use of narrow-spectrum antimicrobial agents. However, the use of narrow-spectrum antimicrobials is low, and that of broad-spectrum antimicrobials is still high. Muraki et al. reported similar data from a nationwide investigation [12]. Since the focus is on the number of antimicrobial prescriptions and the amount of antimicrobial use, it will be necessary to monitor whether the average recovery cycle of patients lengthens in the future while antimicrobial prescriptions decrease, specifically with regard to the percentage of patients who have become severely ill. Antimicrobial prescriptions should be reduced, the patients who need antimicrobials should be prescribed them, and the choice of antimicrobials should be appropriate.

The number of antimicrobial prescriptions for infants and young children has decreased since 2018. The DPY of pediatric pharmacies decreased significantly in all cases. Thus, the decline in antimicrobial prescriptions for infants and young children has been influenced by the decline in DDDs. This is assumed to be largely due to the additional support provided for the appropriate use of pediatric antimicrobials, which was initiated in 2018 [19,20]. However, since pediatric doses are converted to body weight, the DDDs are smaller, and so are the DPY values. Therefore, decreases in the DPY not only occurred due to the added support for the proper use of pediatric antimicrobials, but also due to the influence of DDDs. On the other hand, amoxicillin or amoxicillin–clavulanic acid has been recommended as a first-line drug for acute otitis media in children, according to clinical practice guidelines for the diagnosis and management of acute otitis media in children, updated in 2018 [21]. Therefore, the prescription rate of fluoroquinolones has significantly decreased in pediatric clinics. In contrast, the prescription rates of macrolides remain high in these clinics, even after the implementation of the appropriate use of antimicrobials. The prescription status of macrolides during the study period was confirmed to be mostly clarithromycin followed by azithromycin. Macrolides are well-known anti-inflammatory agents [22,23,24]; therefore, long-term macrolide therapy should be considered to prevent the emergence of antimicrobial-resistant bacteria. If antimicrobial agents are used judiciously in the region, the emergence of AMR and the detection rate of resistant bacteria will decrease. We also believe that by examining the annual trends in antimicrobial use in the community, trends in the detection of antimicrobial-resistant bacteria and characteristics of AMR may emerge. Based on this information, the appropriate use of antimicrobial agents can be promoted. Previous studies have examined DPM as an indicator of antimicrobial use between January 2019 and June 2020 [12]. The DPY defined in this study as more suitable for observing continuous changes in prescriptions. Therefore, it can be stated that DPM is suitable as an indicator of intermittent progress, while the DPY can be used as an indicator of continuous progress.

Reducing the use of antimicrobials may impair critically important treatments. A clinic version of Japan Surveillance for Infection Prevention and Healthcare Epidemiology (J-SIPHE), as part of the Online Monitoring System for Antimicrobial Stewardship at Clinics (OASCIS), has been developed for local clinics, with registrations beginning in June 2022. The first task is to disseminate this system to all clinics. By collecting data on disease names and antimicrobial agents, it is expected to not only prevent inappropriate antimicrobial use but also promote the proper use of antimicrobial agents by physicians. In addition, a community pharmacy version of the J-SIPHE should be developed for the collection of antimicrobial use data from local community pharmacies. By providing feedback on antimicrobial use data to community pharmacies, it is expected that community pharmacists will promote the proper use of antimicrobial agents and contribute to patient awareness.

This study has some limitations. First, we only analyzed prescription data; thus, the actual names of the diseases and the purpose of the prescriptions could not be analyzed. Second, because the 42 pharmacies in this study were located in a limited region of Japan, our data did not reflect the conditions in all regions of Japan. Third, we could not evaluate and compare the DPY with previous reports because this indicator was defined in this study for the first time. Fourth, because of the differences between the place where the antimicrobial agents were prescribed and the place where the patients reside, it was difficult to accurately identify antimicrobial use in the community [25]. However, this study suggests that the AMR Action Plan is effective, to a certain extent, in the studied region and reiterates the importance of appropriate antimicrobial use not only in hospitals but also in the community. We believe that our findings will enable the evaluation of infection control and improve the accuracy of prescribing antimicrobial agents in the community. The newly released AMR Action Plan includes a mention of conducting a study on antimicrobial use starting in pharmacies [26]. This study fits into the part of the plan that proposes a study on the actual status of antimicrobial use. It is necessary for insurance pharmacies and pharmacists to continue to participate in community surveillance to clarify the actual status of antimicrobial use. The study used actual antimicrobial prescriptions dispensed, suggesting that community pharmacists must intervene in the proper use of antimicrobials. Unfortunately, we did not investigate the change in the antimicrobial resistance rate of bacteria isolated in the tested area. The outcomes of the national AMR Action Plan reported a decrease in antimicrobial use, while no change in the isolation rate of antimicrobial-resistant bacteria was found. There may be a time lag between the reduction in antimicrobial use and the isolation rate of antimicrobial-resistant bacteria. Therefore, the relationship between appropriate antimicrobial use and the isolation rate of antimicrobial-resistant bacteria needs to be clarified in the future.

## 5. Conclusions

Our data suggest that the AMR Action Plan and additional support for the appropriate use of antimicrobial agents in children influenced the number of antimicrobial prescriptions in community pharmacies in Japan.

## Figures and Tables

**Figure 1 antibiotics-12-01325-f001:**
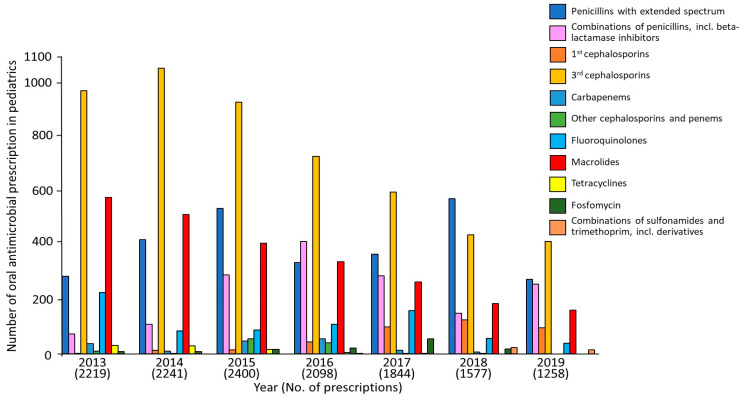
Number of antimicrobial prescriptions between April 2013 and December 2019 in 12 pharmacies located in front of pediatric clinics.

**Table 1 antibiotics-12-01325-t001:** Number of antimicrobial prescriptions between April 2013 and March 2020 in 42 pharmacies.

Antimicrobial Class	Number of Prescriptions in Each Year							
	2013 From April	2014	2015	2016	2017	2018	2019	Total
**AWaRe (Access)**								
Penicillins with extended spectrum	3269	4776	5034	5056	4813	5551	5121	33,620
Combinations of penicillins, incl. beta-lactamase inhibitors	772	1109	1361	1736	1491	998	1089	8556
1st cephalosporins	1019	1210	1181	1226	1074	1397	1182	8289
Combinations of sulfonamides and trimethoprim, incl. derivatives	219	354	441	495	603	733	847	3692
Lincomycins	52	78	50	51	38	49	39	357
**AWaRe (Watch)**								
3rd cephalosporins	28,742	36,632	37,459	32,354	27,686	24,238	21,969	209,080
Macrolides	18,292	25,449	23,537	22,092	17,545	15,869	15,178	137,962
Fluoroquinolones	12,043	15,981	16,699	16,034	15,051	14,849	13,817	104,474
Fosfomycin	796	841	748	655	594	514	432	4580
Carbapenems	193	45	245	261	136	43	17	940
2nd cephalosporins	62	45	68	69	48	4	5	301
Aminoglycoside	26	48	39	22	9	0	0	144
**AWaRe (Access and Watch)**								
Tetracyclines	989	1151	1460	1693	1450	1256	1256	9255
**AWaRe (Reserve)**								
Other cephalosporins and penems	166	109	315	230	190	199	177	1386
**AWaRe (Watch and Not Classified)**								
Anti-tuberculous drug	135	287	245	265	269	263	287	1751
**AWaRe (Not Classified)**								
Combinations for the eradication of *Helicobacter pylori*	359	610	446	483	462	384	354	3098
Other antimicrobials	0	1	1	2	2	8	10	24
Total	67,134	88,726	89,329	82,724	71,461	66,355	61,780	527,509

**Table 2 antibiotics-12-01325-t002:** Annual DPYs of oral antimicrobial agents in 17 representative pharmacies.

Antimicrobial Class	DPY per Year							*p*-Value *
	2013	2014	2015	2016	2017	2018	2019	
**AWaRe (Watch)**								
3rd cephalosporins	157.2	139.5	149.7	129.2	110.8	103.7	98.8	0.02
Fluoroquinolones	110.6	123.5	125.5	138.9	143.3	146.7	148.9	0.04
Macrolides	178.7	176.9	179.2	175.3	157.2	135.3	125.4	0.02

* 2014 to 2016 vs. 2017 to 2019.

**Table 3 antibiotics-12-01325-t003:** Annual DPYs of oral antimicrobial agents in five pharmacies located in front of pediatric clinics.

Antimicrobial Class	DPY per Year							*p*-Value *
	2013	2014	2015	2016	2017	2018	2019	
**AWaRe (Watch)**								
3rd cephalosporins	275.8	263.1	282.9	226.4	185.6	183.0	161.7	0.02
Fluoroquinolones	187.7	116.5	134.6	123.1	107.7	101.8	105.7	0.02
Macrolides	464.2	393.3	393.2	371.3	272.6	260.4	226.7	0.02

* 2014 to 2016 vs. 2017 to 2019.

## Data Availability

Not applicable.

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
