# Peer review of "Surveillance of Antimicrobial Prescriptions in Community Pharmacies Located in Tokyo, Japan"

_antibiotics, 2023, doi:10.3390/antibiotics12081325_

Round 1

Reviewer 1 Report

The reviewer has questions and comments regarding some points in the manuscript. These questions are mainly related to the design of the study and in no way detract from the dignity of this manuscript.

The action plan to combat antimicrobial resistance was only launched in 2016. Why did the authors collect data for such a long period of time (2013-2020)? Maybe in the description of the results and the section "Discussion" it is necessary to compare the periods before 2016 - after 2016? Moreover, the authors point to a noticeable decrease in the number of antibiotic prescriptions in 2018, that is, after 2016.

Another issue also concerns study design. Based on the contents of the manuscript, an antimicrobial resistance action plan is needed to prevent the spread of antimicrobial resistant organisms. The authors noted a decline in antibiotic prescribing since 2018. How has this affected the spread of antibiotic resistance? Maybe it would be worthwhile in the "Discussion" section to somehow consider this?

Author Response

Reviewer 1:

The action plan to combat antimicrobial resistance was only launched in 2016. Why did the authors collect data for such a long period of time (2013-2020)? Maybe in the description of the results and the section "Discussion" it is necessary to compare the periods before 2016 - after 2016? Moreover, the authors point to a noticeable decrease in the number of antibiotic prescriptions in 2018, that is, after 2016.

We appreciate your feedback. According to your comments, the sentences have been revised as below.
Page 5, Lines 184 to 194:
The present study showed a potential effect of the AMR Action Plan on reducing the number of antimicrobial prescriptions in 42 community pharmacies in the Tama area of Tokyo. To the best of our knowledge, this is the first study of antimicrobial prescribing in a limited number of community pharmacies. In this study, we compared the AMR Action Plan in the three years before and after to see its impact. Our data clearly showed a downward trend in the number of antimicrobial prescriptions since 2016. The number of antimicrobial prescriptions as a percentage of total prescriptions has remained at around 10% each year, which may be due to the AMR Action Plan and the additional support for appropriate use of pediatric antimicrobials. In addition, this study was not affected by the COVID-19 pandemic, as data were collected through December 2019, before the pandemic began in Japan.

Another issue also concerns study design. Based on the contents of the manuscript, an antimicrobial resistance action plan is needed to prevent the spread of antimicrobial resistant organisms. The authors noted a decline in antibiotic prescribing since 2018. How has this affected the spread of antibiotic resistance? Maybe it would be worthwhile in the "Discussion" section to somehow consider this?

Thank you for your comments. Unfortunately, we did not investigate the change in the antimicrobial resistance rate of bacteria isolated in the tested area. The outcomes of the national AMR Action Plan reported a decrease in antimicrobial use, while no change in the isolation rate of antimicrobial-resistant bacteria was found. There may be a time lag between the reduction in antimicrobial use and the isolation rate of antimicrobial-resistant bacteria. Therefore, the relationship between appropriate antimicrobial use and the isolation rate of antimicrobial-resistant bacteria needs to be clarified in the future. The sentences have been added in Page 7, Lines 301 to 307.

Reviewer 2 Report

Overall the study provides valuable insights into antibiotic use in the community. I have two questions:

1. What alternatives do physicians use to treat patients with bacterial infections in the face of reduced antibiotic use.

2. Has the average recovery cycle of patients been lengthened by the reduced use of antibiotics, and has there been an increase in the proportion of patients who have become critically ill as a result of the reduced use of antibiotics?

line 45: 'outpatients in Japan is higher' should be 'outpatients in Japan are higher' to correct subject-verb agreement

line 98: Add a comma before 'consisted of'

Author Response

Reviewer 2:
Overall the study provides valuable insights into antibiotic use in the community. I have two questions:
What alternatives do physicians use to treat patients with bacterial infections in the face of reduced antibiotic use.

Since the focus is on the number of antimicrobial prescriptions and the amount of antimicrobial use, it will be necessary in the future to monitor whether the average recovery cycle of patients has lengthened as antimicrobial prescriptions have decreased and with regard to the percentage of patients who have become severely ill. Not only should antimicrobial prescriptions be reduced, but those patients who need antimicrobials should be prescribed them, but the choice of antimicrobials must be appropriate.

Has the average recovery cycle of patients been lengthened by the reduced use of antibiotics, and has there been an increase in the proportion of patients who have become critically ill as a result of the reduced use of antibiotics?

In this study, we did not address patient information and therefore could not examine the impact of reduced antimicrobial prescribing on patient condition. We believe that the relationship between antimicrobial use and patient outcomes needs to be clarified in the future.

line 45: 'outpatients in Japan is higher' should be 'outpatients in Japan are higher' to correct subject-verb agreement

Thank you for your suggestion. We have corrected it.

line 98: Add a comma before 'consisted of'

Thank you for your suggestion. We have corrected it.

Reviewer 3 Report

This study is generally interesting e.g. to the readers that would need to compare the similar type of data in country other than Japan.

Actually, I do not have any particular critical remark related to the presented paper. It was rather well designed (it would of course benefit from adding more information about the diseases that the antimicrobials were prescribed to, but I understand that since the study is finished, this is not longer posible and was also addressed in one of the final paragraphs).

I think that adding a short summary or conclusion section would make it more clear to the readers.

Author Response

Reviewer 3:
This study is generally interesting e.g. to the readers that would need to compare the similar type of data in country other than Japan.
Actually, I do not have any particular critical remark related to the presented paper. It was rather well designed (it would of course benefit from adding more information about the diseases that the antimicrobials were prescribed to, but I understand that since the study is finished, this is not longer posible and was also addressed in one of the final paragraphs).
I think that adding a short summary or conclusion section would make it more clear to the readers.

We appreciate your comments and suggestions. We have added the Conclusion section as below.
Page 8, Lines 310 to 313:
5. Conclusion.
Our data suggest that the AMR Action Plan and additional support for the appropriate use of antimicrobial agents in children influenced the number of antimicrobial prescrip-tions in community pharmacies in Japan.